# Physiological and Physical Strategies to Minimize Damage at the Branch–Stem Junction of Trees: Using the Finite Element Method to Analyze Stress in Four Branch–Stem Features

**DOI:** 10.3390/plants12234060

**Published:** 2023-12-03

**Authors:** Tung-Chi Liu, Yi-Sen Peng, Bai-You Cheng

**Affiliations:** 1Department of Horticulture, National Chung Hsing University, Taichung City 40227, Taiwan; liuok@dragon.nchu.edu.tw (T.-C.L.); eason.peng@smail.nchu.edu.tw (Y.-S.P.); 2Graduate Institute of Environmental Resources Management, TransWorld University, Douliu City 64063, Taiwan

**Keywords:** branch–stem structure, branch collar, branch bark ridge, finite element method, mechanical function, thickening of the lower stem

## Abstract

This study analyzed the mechanical and physiological strategies associated with four features in the branch–stem junction of a tree, namely the U-shaped branch attachment, the branch collar, the branch bark ridge, and the roughened lower stem. Models were established for each stage of tree growth by adding these four features sequentially to a base model, and the finite element method (FEM) was employed to create three-dimensional models of an Acer tree’s branch–stem structure for static force analysis. According to the results, the development of the branch collar shifted the point of breakage to the outer part of the collar and, thus, constituted a physiological strategy that prevented decay in the stem. Additionally, the concentration of stress in the branch bark ridge limited the area of tear in the bark following breakage. Finally, the U-shaped branch attachment reduced stress and shifted the point of peak stress toward the branch, while the thickening of the lower stem reduced the overall stress. The development of these features, including the spatial positioning of the branch bark ridge and branch collar, resulted in two breakage points constituting a physical and a physiological strategy that limited damage to the tree and protected the xylem structure. This is the part that has been challenging to decipher in previous discussions of tree-related self-protection mechanisms.

## 1. Introduction

Taiwan is susceptible to tropical cyclones in the summer, which can cause serious damages to street trees in cities and compromise human safety due to the destruction of trees [1,2]. Regarding such damage, a common location of branch breakage is the junction of the branch and stem [3,4,5,6,7].

In a tree, a branch–stem structure is the connecting structure of two branches with different diameters. Studies have observed the morphological effects of the branch–stem junction on a tree’s strength [8,9,10,11,12], with the assessment being based on the analysis of Shigo’s anatomical findings on the tree’s branch–stem structure [5], which defines the following four common morphologies of wood fiber during its growth, as shown in Figure 1:U-shaped branch attachment: the connection between a branch and a stem forms a smooth U-shaped curve as a result of the transition angle at the connection between the cylindrical branch and the stem [3,13,14].Branch collar: The branch collar is the location where the xylem tissues of the branch and stem overlap in secondary growth [15,16]. The structural transition between the branch and stem at this junction exhibits a gradual and smooth curvature, with a observable dividing line between the branch and stem [6,15].Branch bark ridge: The ridge is a raised area formed on the bark where the branch and stem meet as a result of compression between the bark tissues of the branch and stem. The ridge extends downward to the transitional area between the branch and stem [8,17].Thickening of the lower stem: this thickening occurs at the bottom of the junction between the branch and stem, where the stem thickens as part of secondary growth [10].

**Figure 1 plants-12-04060-f001:**
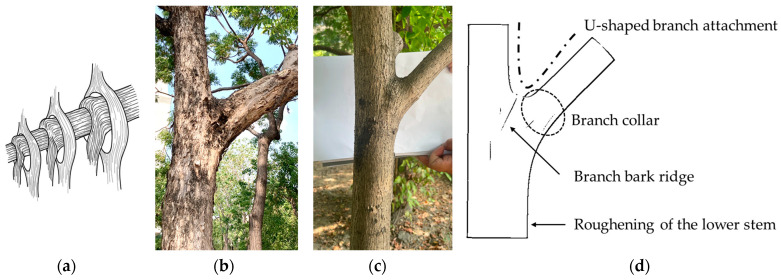
Branch–stem structure. (**a**) Explanation of branch and stem growth according to Shigo’s model. (**b**,**c**) Photographs of *Bischofia javanica* and *Hibiscus tiliaceus*, respectively. (**d**) Illustration of the branch–stem structure along with the four distinct features.

The morphology of the branch–stem structure has been verified as a critical contributing factor of the branch–stem junction’s strength [8,9]; however, most related studies have been limited to statistical investigations regarding correlations among factors. Shigo’s branch–stem model, despite explaining the transportation of water and nutrients, fails to address both mechanical support and mechanical connection [5,18]. Furthermore, research exploring the morphological effects of the branch–stem structure on mechanical behavior has predominantly measured the parameters of partial characteristics, such as the diameter ratio, angle, and junction area, for statistical investigations, with few discussions on the overall effects of morphology on the junction’s strength [8,11,19]. Some studies have developed equations to explain the mechanical behavior of the branch–stem junction; however, the oversimplification of these equations has led to the most critical contributing factor, namely morphological changes in the junction, being overlooked [20]. For a deeper understanding of how this factor and other characteristics affect the structure and mechanical mechanism of trees, the present study employed the finite element method (FEM) to deconstruct, model, and compare the morphologies of actual trees, with a view toward accurately evaluating the functional differences among morphologically different branch–stem junctions.

## 2. Results

### 2.1. Results Regarding Individual Morphologies

To reveal morphological differences, the morphologies of multiple branch–stem junctions were divided into five stages (hereinafter referred to as “PT1”–“PT5” to denote patterns) according to Shigo’s model, and the FEM was used for a static analysis to obtain the von Mises stress of each model group (Figure 2). As von Mises stress represents a combined stress, there are no negative values. The magnitude of stress is indicated by colors: red areas denote high stress, while blue areas indicate lower stress levels. Stress levels are color-coded: red for high stress and blue for lower stress. The images are organized from PT1 to PT5, with the rows sorted by angles: 45°, 60°, and 75°.

Additionally, the stress at the crotch angles is specifically illustrated (Figure 3). The layout of these images follows the same pattern as Figure 2.

### 2.2. Comprehensive Comparison

To clarify the relationship between stress distribution and morphology, this study juxtaposed the results of individual model groups to visualize how the concentrated stress changed and moved between the branch and the stem (Figure 4 and Figure 5). In Figure 5a, areas with concentrated stress are shown in red and yellow, and those with low stress are shown in blue. The distribution of changes is clearly visible in the figures. Figure 5b presents the stress results in the form of a stress–position graph; the maximum stress value moved substantially outward toward the branch between PT1 and PT2 and again between PT2 and PT3 before it started moving slightly inward between PT3 and PT4 and again between PT4 and PT5. Finally, the maximum stress of PT5 fell at a position between those of PT2 and PT3, slightly nearer to that of PT3. In Figure 5b, PT1–PT5 are marked in red, blue, green, purple, and yellow, respectively, and the dotted lines indicate the locations of stress concentration shown in Figure 5a.

### 2.3. Analysis of Changes in Crotch Angle

The stress gradient changed slightly with changes in the crotch angle, albeit to a lesser extent compared to its changes caused by morphologies (Figure 6). Accordingly, changes in the crotch angle did not affect the experimental results of the morphological observations (Figure 7). The finding that the crotch angle does not affect the strength of the branch–stem junctions is consistent with that reported by Kane (2007) [21]. 

## 3. Discussion

Trees exhibit primary and secondary growth. Primary growth occurs at the bud and root tip and determines the arrangement of each part of the tree body, whereas secondary growth determines the thickness of the tree body (Figure 1a). The present simulation revealed that different morphologies resulted in different stress distributions (Figure 4) and possibly different breakage patterns (Figure 6). PT1 had substantial stress concentration in the stem. However, the stress became less concentrated and began moving toward the branch when a U-shaped junction was formed (PT2). In PT3, in which a branch collar was formed, the stress was slightly lower than that in PT2 and moved toward the branch. However, the formation of a branch bark ridge in PT3 caused the stress to shift slightly toward the stem and become concentrated as a result of the raised structure, with a stress increase in the area between the branch bark ridge and the stem. Finally, in PT5, in which the stem roughened, no movement of the concentrated stress was observed, although the maximum concentrated stress decreased.

Although studies have reported that a high-stress site grows rapidly to homogenize the stress distribution [22], the present study’s analysis revealed that the growth progression through the four morphologies did not necessarily homogenize the stress distribution. For example, the growth of a branch bark ridge in PT4 (Figure 5a) further concentrated the stress. Given that this growth could not be explained by the need to homogenize stress distribution, it might have provided a different morphological function that warrants further investigation. Meanwhile, the position of stress concentration shifted toward the branch as additional parts were developed in the tree (PT3 and PT4). This movement of stress concentration maintained the location of breakage, if there was any, farther away from the stem, leaving the vertical stem fibers intact and facilitating the compartmentation of decay [15]. In other words, keeping a breakage outside the area of the branch collar can uphold the integrity of the reaction zone [23]. Previous research has established the branch collar as a significant indicator of decay patterns in trees [24]. While these studies have identified a correlation, the present study extends this understanding by providing a visual explanation of the branch collar’s role in physiological protection. This approach enhances the comprehension of its underlying mechanics and implications for tree health. The findings herein corroborate the structural significance of the branch collar observed in our research, particularly in terms of stress distribution and its management in trees. By integrating physiological and structural perspectives, this study underscores the critical role of the branch collar as a morphological feature essential for the maintenance and health of trees.

Conversely, stress concentration near the stem may cause breakage between the branch and the stem, aligning with the stem; such damage is difficult for a tree to recover from [15,25]. Accordingly, in the discussion regarding morphological functions, the normal physiological functioning of a tree may result from the tree purposely causing structural defects in certain parts of its body to prevent more severe damage to parts outside a branch–stem junction [12]. Additionally, no branch collar or swelling at the base of the branch in PT1 or PT5 (Figure 5b) resulted in high stress or stress concentration near the stem. This finding was consistent with the observations during a field survey regarding thick dead branches and wilt trees [25]; therefore, the physical structures in PT1 and PT2, particularly that in PT1, were at risk of breakage and demonstrated signs of deterioration.

The U-shaped branch attachment and thickening of the lower stem are two distinct geometric features in the branch–stem structure of a tree, and both result from stress optimization. The U-shaped branch attachment may be a natural result of structural optimization that enables stress to be distributed uniformly along the branch and stem [26,27]. The decrease in stress, particularly in areas of stress concentration, is likely a result of the stem’s thickening. This indicates that changes in morphology, even in non-stress concentrated areas, can also play a role in mitigating risks. According to Figure 5 and Figure 6, the U-shaped branch attachment yielded noticeable changes in the stress–position relationship; the thickening of the lower stem possibly improved the overall stability and durability of the stress while reducing the risks of breakage and damage.

The branch collar and branch bark ridge may have functioned to guide the position of stress concentration. Specifically, the branch collar may have provided some protection that leads to a change in the position of stress concentration, whereas the branch bark ridge possibly contributes to the control and balancing of stress distribution. These features may have enhanced the protection of the transportation in the outer layer of trees and achieved balance among multiple survival strategies. It is noteworthy that trees are primarily composed of two types of materials: the softer outer bark and the harder wood at the center. The branch bark ridge and branch collar are, respectively, formed from these two materials. Therefore, while both features lead to concentrated stress points offering protective benefits, their material composition suggests they may yield different outcomes. When a branch breaks, the break in the wood and bark might differ due to the mechanical properties of these materials. This implies that trees effectively ‘set up’ two distinct damage control points using these different material-based features. The branch bark ridge controls the endpoint of bark tearing, preventing further damage to the main trunk’s bark, while the branch collar manages the breakage of side branches to avoid harm to the main trunk’s fibers. Further, the former protects the integrity of the transport system, while the latter safeguards the structural system.

Finally, changes in the crotch angle revealed the propensity of the stress gradient to change alongside morphological changes (Figure 7). This observation verified the more critical role of morphological characteristics in tree management compared to the crotch angle; this finding was consistent with the argument of multiple studies that the crotch angle has a limited effect on the strength of a branch–stem junction [11,19]. The simulation result regarding force exertion revealed that the stress distribution remained unchanged under multiple degrees of force within the linear elastic range.

## 4. Materials and Methods

This section begins with an introduction to the FEM. This introduction is followed by discussions regarding the sources of materials, the finalization of the model, the setting of the simulation environment, and the selection of experimental methods, resulting in a total of five subsections. The arrangement of these subsections clearly guides readers through each step of the research and, thus, helps them understand the context of this study.

### 4.1. FEM

The purpose of this study was to understand the mechanical functions of branch and stem features in a tree. Given that the morphological complexity involved in a branch–stem junction precludes the use of analytical methods, the FEM, a commonly adopted method for solid mechanics analyses, was considered an appropriate option for solving the physical problem at hand.

The FEM involves dividing a physical problem into multiple smaller finite elements for separate calculation; the calculation results related to these individual elements are then used to infer an answer to the overall problem. This method stemmed from the idea of discretizing continuous infinite variables, which was proposed by the mathematician Richard Courant in 1943 in his work entitled *Variational Methods for the Solution of Problems of Equilibrium and Vibrations* [28]. Since the 1980s, the FEM has been incorporated into multiple commercial software packages (e.g., Ansys (Version 2023 R2), Abaqus (Version 2023), COMSOL Multiphysics (Version 6.1), and SOLIDWORKS (Version 2023)) as their main numerical analysis method.

The extensive use of the FEM in research on tree mechanics is proof of the method’s effectiveness. Mattheck employed the FEM to calculate the relationship between tree morphology and stress distribution [22,26,27]. Ahmadi et al. analyzed the dynamic behavior of an apple in collision by using the FEM [29]. Jackson et al. adopted the FEM to simulate the three-dimensional geometric shapes of 21 trees and to predict the degrees of mechanical strain on those trees’ stems [30]. Tsugawa et al. employed a lidar scanner to obtain the three-dimensional point cloud of Zelkova serrata and Larix kaempferi and extracted the cylinder structure in the point cloud for FEM simulations to evaluate the mechanical stress and other mechanical characteristics in response to gravity [31].

The aforementioned studies verified the high performance of the FEM in analyzing and simulating physical problems, particularly in research regarding the branch–stem structure. In summary, the FEM facilitates the precise presentation of shape, the simulation of diverse morphologies, the visualization of overall mechanisms, and the detailed analysis of each tree part. Accordingly, the FEM was adopted in the present study to investigate characteristic differences in the branch–stem junction.

### 4.2. Experimental Materials 

The specimens utilized in this experiment were sourced from campus trees located in Taichung, a city in central Taiwan. Acer trees were selected as the subject of study due to their widespread distribution in temperate and subtropical areas (Figure 8a). They are representative of the general response of tree species in this region under the impact of severe wind forces, making them a pertinent choice for studying branch–stem junction models.

To acquire precise data on the tree branch–stem structure, we utilized a Revopoint POP 2 scanner (Revopoint 3D Technologies Inc., Xi’an and Shenzhen, China). The main advantages of this scanner are its portability and ease of handheld operation, enabling accurate on-site scanning.

The scanning process generated point cloud data of the branch–stem structure, which was then processed using the Revo Scan 5 software, Version 5.3.3 (as updated on 2023-11-16 for Windows and Mac) provided by the same manufacturer. The processing steps included transforming the point cloud data into a continuous surface. This transformation created a thin shell (surface) model, which was crucial for subsequent modeling. These processing steps ensured the accuracy and completeness of the created 3D models, providing a precise reference for the establishment of the tree’s branch–stem structure (Figure 8b,c).

### 4.3. Preprocessing

The processing stage involved the model setting and preparatory procedures, including defining the geometric shape, material properties, and boundary conditions, as well as meshing. This step provided a basis for subsequent analyses and calculations.

#### 4.3.1. Defining the Geometric Shape: Modeling of Branch–Stem Structures 

On the basis of Shigo’s branch–stem model [5], the present study defined four morphological features of growth (Figure 9): the U-shaped branch attachment, the branch collar, the branch bark ridge, and the thickening of the lower stem, hereinafter coded as a, b, c, and d, respectively.

Model groups were built by first establishing a base model. Given that the aforementioned morphological features were developed alongside the growth of the tree, they were added to the base model sequentially to form four models—namely models with features a, a + b, a + b + c, and a + b + c + d—in addition to the base model. For clarity, the base model is hereinafter referred to as PT1, and the other four models are referred to as PT2–PT5, respectively (Figure 10). The crotch angle of the branch material used in this experiment was 45°. However, to address the variability in the crotch angle, this study investigated the crotch angles of 62 *Acer* trees in an urban area in central Taiwan and found that the crotch angles of these trees exhibited a normal distribution (Figure 11), with a mean of 46.2 and a standard deviation of 17.7. Therefore, models with 60° and 75° crotch angles were established, with all other features being kept constant (Figure 10), to reveal differences that were attributable to the crotch angle.

#### 4.3.2. Defining Material Properties 

The material parameters were defined based on the Acer material data from the software’s database. The parameters included elastic modulus, Poisson’s ratio, shear modulus, mass density, and yield strength, which are summarized in Table 1. The yield strength, while essential for finite element simulation software, was assumed to be within a specified range of the stress intensity simulated in this study. Therefore, it does not affect the results discussed in this paper. Following the approach of Dounar and Iakimovitch [32], the present study regarded the material properties as isotropic and as conforming to Hooke’s law, despite the nonuniform and anisotropic nature of wood materials, to facilitate calculation and to focus on the effects of morphological features on stress. 

#### 4.3.3. Setting Boundary Conditions and Meshing

Figure 12a presents the boundary condition setting on the mesh. To ensure that the established mesh accommodated a sufficient quantity of appropriately small finite elements, this study set the number of Jacobian points to 16 to create a high-quality mesh. Because overly flat elements can lead to substantial errors in processing during problem solving, this study ensured that the experiment did not contain any distorted finite elements or elements with a height/width ratio higher than 1:10. 

A uniform vertical downward force of 100 Newtons (N) was applied on the plane of the branch’s top end, relative to the direction of the branch’s extension, to simulate the effect of wind loads (F1), as depicted in Figure 12b. This force, acting on a circular surface with a diameter of 0.2 m, generated a pressure of approximately 3183 Pa. The free end and fixed end were set at the stem’s top and bottom surface, respectively.

#### 4.3.4. Simulation Setup

A static force analysis was performed using SOLIDWORKS^®^ Simulation 2020 (Dassault Systèmes SolidWorks Corporation, Waltham, MA, USA). When deformation-induced changes in the direction of force exertion are not considered, the stress concentration coefficient depends entirely on the structure [33]. In this experiment, von Mises stress was employed as the basis for comparing the stress distribution results of the models.

Given that the stress distribution was symmetrical with respect to the cross section, the plane of symmetry at the crotch was used to map the stress–position curve, namely the location along the red arrow in Figure 12b. The x-axis was normalized (i.e., X0 (m)) to compare different patterns with one another. Our study focused on understanding the impact of tree morphology on stress changes. This led us to analyze the material behavior within the elastic limits. This approach allows a focused examination of how tree morphology influences stress distribution.

### 4.4. Experimental Limits

With Acer trees as its subjects, the present experiment revealed universal principles and mechanisms generalizable to all tree species through the use of the FEM and related analyses, aiming to transcend material variations like tree species. We specifically chose stress levels to ensure that the analysis remains within the elastic range of the materials. However, the differences in growth characteristics and physical properties between species, such as those between species-specific growth models or structural properties, necessitate adjustments or modifications to the proposed models. In addition, this study had the following limitations: (1) The assumption that trees demonstrate isotropic material properties overlooked the anisotropic and nonuniform nature of tree-based materials. (2) There were experimental constraints of using plastic materials to perform simulation. (3) The simulation did not consider environmental loads other than wind, such as rain or snow. Despite the favorable performance of the proposed models in relation to Acer trees, the aforementioned limitations must be addressed in future research and should be considered in the interpretation of the present findings.

## 5. Conclusions

This study explored four morphological features of the branch–stem structure in trees to understand the mechanical strategies and effects of these features. Employing the FEM for stress analyses related to the branch–stem junction, we were able to compare various structures comprehensively. Our analysis builds upon existing research by providing, for the first time, a comprehensive visual comparison of stress distribution across these distinct morphological features. While previous studies have visualized stress in specific scenarios, our research delves deeper, examining the growth progression of these morphologies and their potential impact on stress distribution. The detailed observation and analysis of the stress–position relationship led to significant findings, offering a new perspective for understanding the complex interplay between morphological characteristics and stress distribution. This approach enriches our knowledge of tree growth and stress response mechanisms, contributing to the broader understanding of tree mechanical strategies. 

For clarification of the mechanical differences among morphologies of the branch–stem junction, this study investigated the functions of multiple morphologies, namely the U-shaped branch attachment, branch collar, branch bark ridge, and thickening of the lower stem. This investigation revealed how the branch–stem structure of a tree adapts to external conditions.

Geometric changes contributed to not only stress optimization (stress homogenization) but also the guidance of the breakage point. Specific geometric characteristics improved the uniformity of stress distribution and guided concentrated stress toward a specific location to enhance the overall stability and durability of the tree. The morphological features influenced the location of breakage to minimize damage to tissue systems. Through simulation analysis, this study aimed to elucidate the mechanical mechanisms induced by the presence of four distinct features by comparing each experimental sample with the addition of only one feature each time, thereby clarifying the function of each. Additionally, the experiment varied the angles of these features and applied forces in different directions to assess their functional robustness. The comparison of morphologies suggested that the location of stress concentration is related to specific biomechanical strategies, which may protect various tissue systems, including the transportation system in the outer layer, and maintain a balance among survival strategies.

Overall, each of the aforementioned four morphological features played a critical role in the formation of the branch–stem junction. Our findings provide new insights into the mechanical properties of a tree’s branch–stem structure and reveal unexpected strategies trees adopt for stress concentration control. This study, integrating interdisciplinary techniques, built a model from mathematical modeling and employed physiological and theoretical aspects to analyze detailed variations. The methodology of this study not only enhanced our understanding of tree risk mitigation mechanisms but also opened avenues for future research, setting the stage for exploring more intricate scenarios. This research lays the groundwork for future studies on the morphological variations of branch-stem structures in different tree species, local material changes, and multidirectional external forces, providing a viable methodological foundation for mechanical interpretation in urban tree risk assessment. These insights could serve as a valuable reference for street trees’ management, urban planning, and ecological protection, contributing to the harmonious coexistence between humans and nature.

## Figures and Tables

**Figure 2 plants-12-04060-f002:**
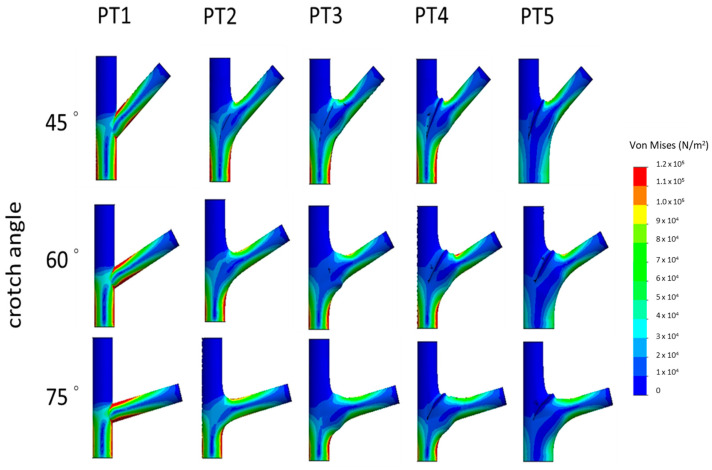
The illustration shows the von Mises stress distribution on the surface of a tree branch.

**Figure 3 plants-12-04060-f003:**
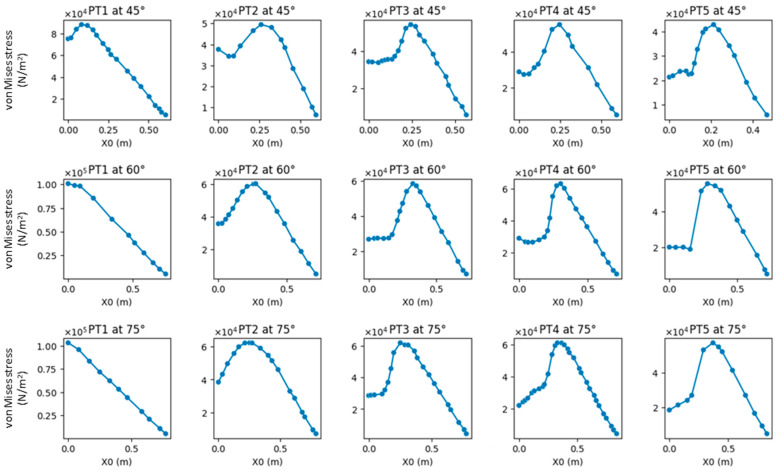
Comparison of stress–position graphs in PT1–PT5.

**Figure 4 plants-12-04060-f004:**
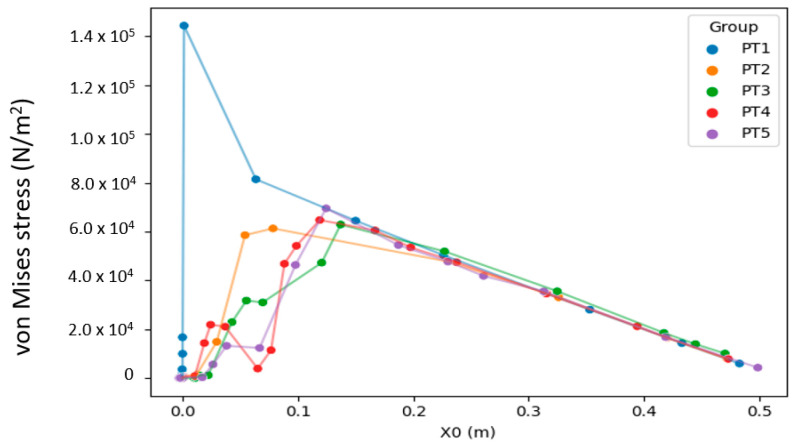
Comparison of stress–position graphs in PT1–PT5 at force F = 3183 Pa and crotch angle α = 45°.

**Figure 5 plants-12-04060-f005:**
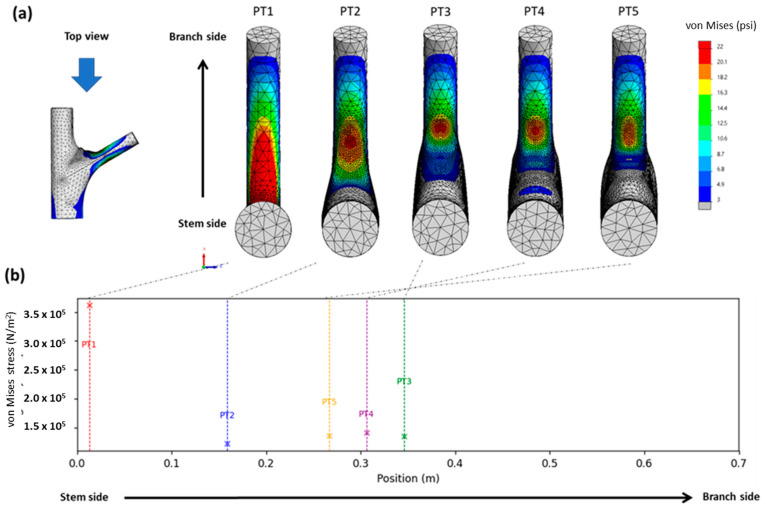
Maximum stress and movement of stress: (**a**) top view of equivalent stress distribution in the branch–stem structure; (**b**) relationship between stress concentration and its position relative to the stem and branch, with PT1–PT5 being marked in red, blue, green, purple, and yellow, respectively, and with dotted lines indicating the locations of stress concentration shown in (**a**).

**Figure 6 plants-12-04060-f006:**
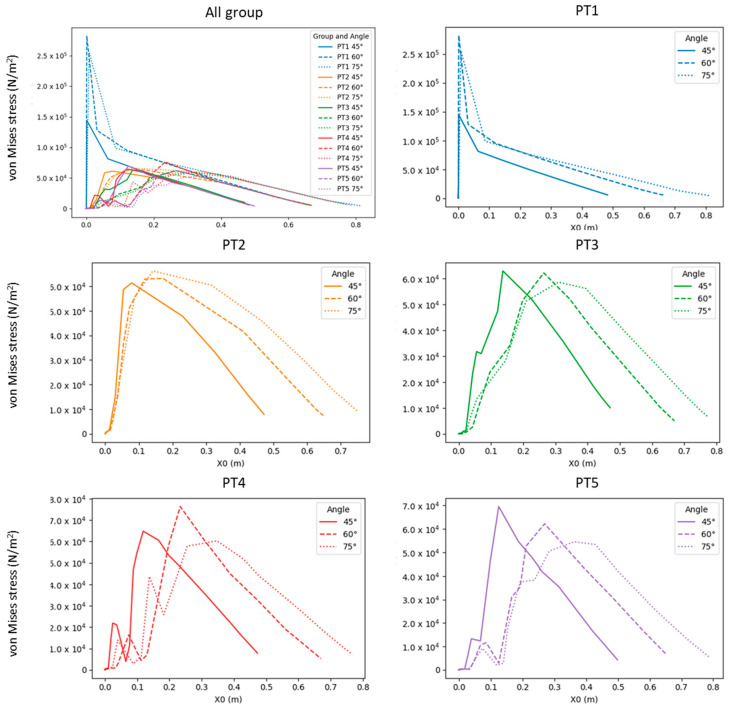
Comparison of stress–position graphs in PT1–PT5 in relation to crotch angle.

**Figure 7 plants-12-04060-f007:**
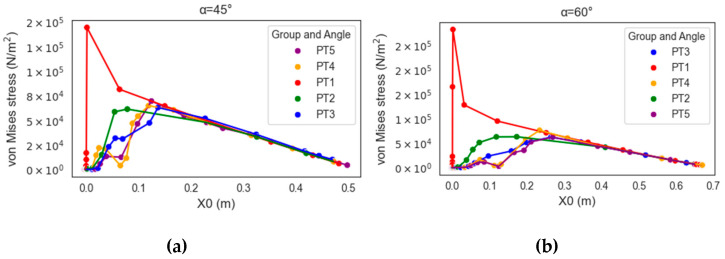
Stress–position graphs in PT1–PT5 depicting panels (**a**–**c**) at crotch angles of 45°, 60°, and 75°, respectively.

**Figure 8 plants-12-04060-f008:**
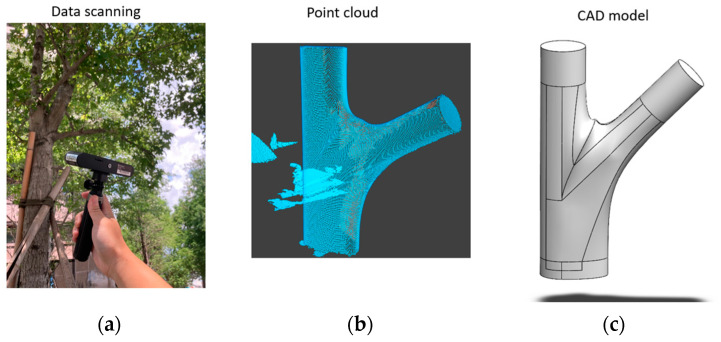
Data collection process for the branch–stem structure: (**a**) scanning a tree, (**b**) point cloud generation, and (**c**) computer-aided design model.

**Figure 9 plants-12-04060-f009:**
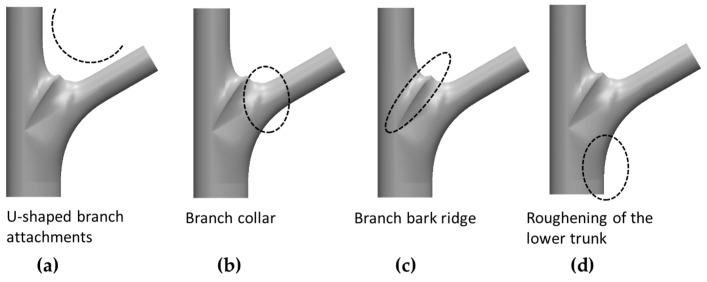
Illustration of the branch-stem structure highlighting the locations of key features: (**a**) U-shaped branch attachment located at the junction, (**b**) branch collar surrounding the base of the branch, (**c**) branch bark ridge along the upper seam of the branch, and (**d**) thickening at the lower part of the stem.

**Figure 10 plants-12-04060-f010:**
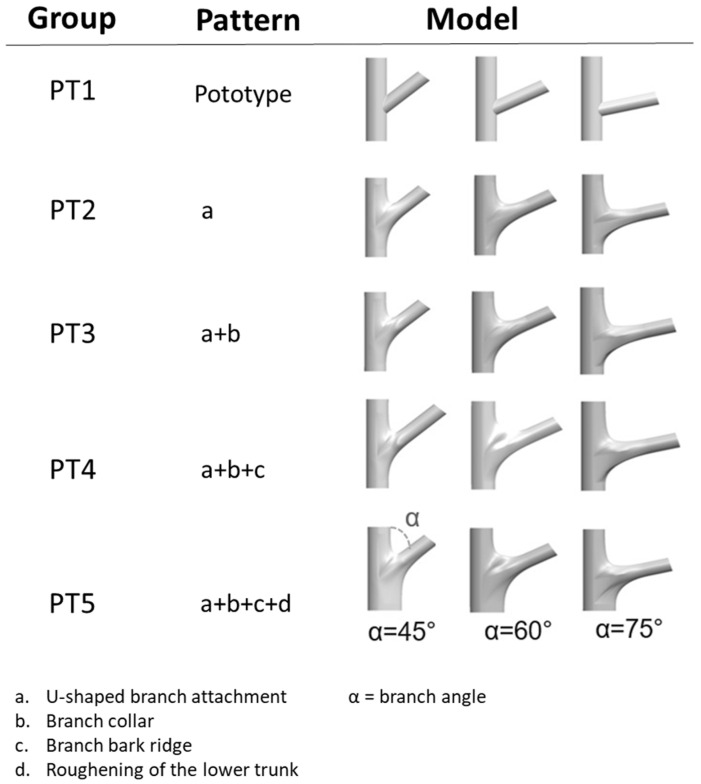
Models for analysis and the progression of feature development. From top to bottom, these models started with a base model (PT1), to which a U-shaped branch attachment (PT2), a branch collar (PT3), a branch bark ridge (PT4), and a roughened lower stem (PT5) were added sequentially. From left to right are the models with crotch angles of 45°, 60°, and 75°.

**Figure 11 plants-12-04060-f011:**
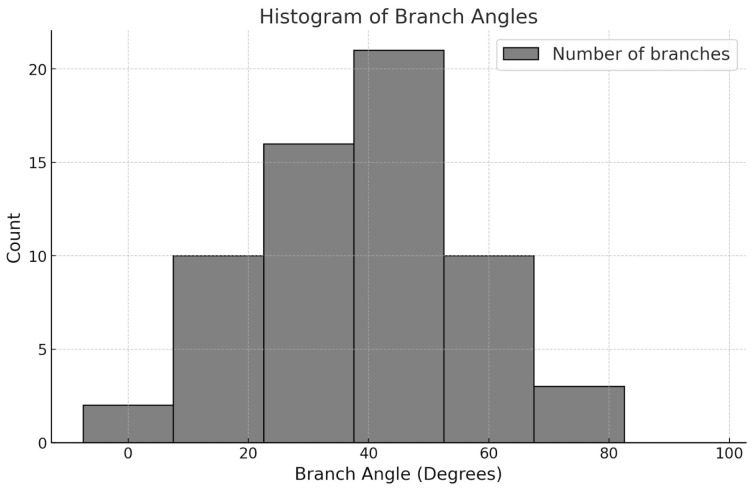
Crotch angle of 62 Acer trees in an urban area in central Taiwan.

**Figure 12 plants-12-04060-f012:**
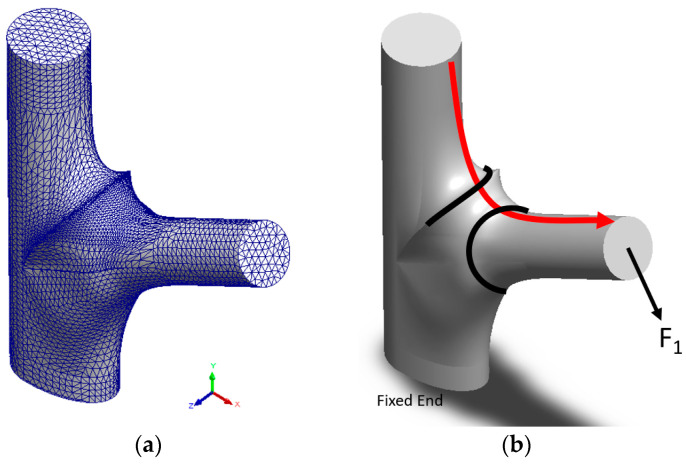
Mesh, area of interest, and boundary conditions: (**a**) shows the appearance of the model after meshing, with the mesh being densified in the area where high stress variation is expected; (**b**) direction and method (F1) of force exertion, fixed end, area of interest (red arrow), and relationship between the branch bark collar and the branch collar.

**Table 1 plants-12-04060-t001:** Material properties of Acer.

Material Property	Value
Elastic modulus	3000 Mpa
Poisson’s ratio	0.29
Shear modulus	300 Mpa
Mass density	159.99

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
