# Peer review of "Physiological and Physical Strategies to Minimize Damage at the Branch–Stem Junction of Trees: Using the Finite Element Method to Analyze Stress in Four Branch–Stem Features"

_plants, 2023, doi:10.3390/plants12234060_

Round 1

Reviewer 1 Report

Comments and Suggestions for Authors

This is an innovative, interesting, and important investigation.

The major issue this reviewer has found is that the type of applied loading has not been properly documented. Line 274 refers to three wind loads in Newtons. However, the direction, as well as the impact point, remain unclear. How is it possible to give the loading in Newtons, whereas all the stresses are given in Pascals?

It appears that all stresses are proportional to the applied load level. Does this not make two of the three loading levels redundant?

This reviewer wonders what kind of bias possibly results from the isotropic von Mises material model. Is it possible to discuss this issue more extensively?

The written expression needs polishing. Wood fiber morphology does not appear relevant in this study?  It appears that the Figures are not numbered in the order of appearance in the text.

Author Response

Dear respected reviewer, please find my response attached in the document.

Reviewer 2 Report

Comments and Suggestions for Authors

This manuscript is a study that utilizes finite element methods to examine the stresses on branch-stem junctions. The authors have divided the morphology of branch-trunk junctions into four sections in Part I.I suggest the authors add a figure under this paragraph, and it would be better if there is a real plant figure. This manuscript does not follow the usual format, with the results being presented in the second part. The authors should make sure that the journal's requirements are fulfilled. Furthermore, the order of the figures is not laid out in order.

Overall, this paper used one model to simulate the morphological structure of four branch and trunk junctions, but no adequate field multi-species surveys were actually carried out, and only lacquer trees were sampled for the surveys. Although the analysis process is more systematic, the findings of the article are too limited. It is not suitable to be published as a research paper, but more like an experimental report. I hope the editor can synthesize the opinions of other reviewers to make a decision. In the field of forestry research, especially about morphology, adequate research in the field is necessary as a basis for a significant study. In addition, the title of this paper is to reduce damage at the branch-trunk junction, but how can the results of this manuscript be used in forest management to achieve this purpose? I think it is very difficult, at least from the current conclusions.

Comments on the Quality of English Language

Minor editing of English language required

Author Response

(The authors gave the same response as above.)

Round 2

Reviewer 1 Report

Comments and Suggestions for Authors

This reviewer has difficulties in understanding the Authors response to the comment “It appears that all stresses are proportional to the applied load level. Does this not make two of the three loading levels redundant?”.

If the stress does vary proportionally with the load, the different load levels are redundant.

However, there is a yield stress mentioned in the material model description. Does this mean that the stress does not vary proportionally with the load?

As there is a yield stress given, there probably also should be a plasticity model (hardening rule, flow rule, etc.). This reviewer was unable to find any description of a plasticity model.

Author Response

Thank you for the reviewer's guidance. Please find our responses in the attached document.

Reviewer 2 Report

Comments and Suggestions for Authors

I have no further major comments for the authors except to refine the conclusion.

Author Response

(The authors gave the same response as above.)

Round 3

Reviewer 1 Report

Comments and Suggestions for Authors

This reviewer still has difficulties understanding the role of the yield stress in the numerical experiments. What happened when the yield stress was reached? Or were all the treatments conducted in the elastic range, so that the yield stress was redundant?

Author Response

Dear reviewer, our response is as attached.

Comments and Suggestions for Authors

Main concerns:

This reviewer still has difficulties understanding the role of the yield stress in the numerical experiments. What happened when the yield stress was reached? Or were all the treatments conducted in the elastic range, so that the yield stress was redundant?

Reply

Thank you for your insightful comments. All procedures in our study were conducted within the realm of elasticity. Firstly, past research has highlighted four key morphological factors related to breakage, and our article focuses on the impact of variations in these morphological features on branch breakage. This aspect is of particular interest to us, more so than other parts of the study.

Yield stress is one of the essential parameters for running finite element simulation software. Although the manuscript accurately presents the yield stress values used in the simulations, it does not affect any of the results discussed in our paper. We appreciate the reviewer’s detailed observation and inquiry.

The corrections we will make include: 1. Eliminating the yield stress texts to avoid unnecessary misunderstandings. 2. Clarifying in the text that while the parameter of yield stress is necessary in our experiments, it does not influence the outcomes discussed in this paper.

Revised lines code

1)     Materials and Methods (page 10, line 296 through line 299).

2)     Materials and Methods (page 11, line 340).
